# Sharing the load: How a personally coloured calculator for grapheme-colour synaesthetes can reduce processing costs

Joshua J. Berger[ID][1]*, Irina M. Harris[2], Karen M. Whittingham[3], Zoe Terpening[2,4], John D. G. Watson[1,4]

1 Faculty of Medicine and Health, The University of Sydney, Sydney, NSW, Australia, 2 School of Psychology, Faculty of Science, The University of Sydney, Sydney, NSW, Australia, 3 Faculty of Science, University of New South Wales, Sydney, NSW, Australia, 4 Faculty of Medicine and Health, University of New South Wales, Sydney, NSW, Australia

* joshua.berger@sydney.edu.au

**Data Availability Statement:** The anonymised data that supports the findings of this study may be found at: https://www.kaggle.com/joshuajamesberger/sharingtheload.

## Abstract

*Synaesthesia* refers to a diverse group of perceptions. These unusual perceptions are defined by the experience of *concurrents;* these are conscious experiences that are catalysed by attention to some normally unrelated stimulus, the *inducer*. In grapheme-colour synaesthesia numbers, letters, and words can all cause colour *concurrents*, and these are independent of the actual colour with which the graphemes are displayed. For example, when seeing the numeral '3' a person with synaesthesia might experience green as the concurrent irrespective of whether the numeral is printed in blue, black, or red. As a trait, synaesthesia has the potential to cause both positive and negative effects. However, regardless of the end effect, synaesthesia incurs an initial cost when compared with its equivalent example from normal perception; this is the additional processing cost needed to generate the information on the *concurrent*. We contend that this cost can be reduced by mirroring the concurrent in the environment. We designed the Digital-Colour Calculator (DCC) app, allowing each user to personalise and select the colours with which it displays its digits; it is the first reported example of a device/approach that leverages the concurrent. In this article we report on the reactions to the DCC for a sample of fifty-three synaesthetes and thirty-five non-synaesthetes. The synaesthetes showed a strong preference for the DCC over its normal counterpart. The non-synaesthetes showed no obvious preference. When using the DCC a subsample of the synaesthete group showed consistent improvement in task speed (around 8%) whereas no synaesthete showed a decrement in their speed.

## Introduction

Synaesthesia is an unusual sort of perception. It is easiest to describe synaesthesia by its contrast to normal perception. In the typical processes of perception, a stimulus is paired with some common experience. For example, light is normally paired with vision and airwaves are

**Funding:** The author(s) received no specific funding for this work.

**Competing interests:** The authors have declared that no competing interests exist.

normally paired with sound. However, with synaesthesia at least one additional experience is also paired with the stimulus, which is termed the *inducer*; the additional experiences are termed the *concurrents* and they are catalysed by attention that is paid to the *inducer* [1–6]. For example, when some people hear music they also see colour: this is music-colour synaesthesia [7]. For a more detailed overview of synaesthesia in general, see *Synesthesia* in Annual Review of Psychology [8].

Grapheme-colour synaesthesia is another type: one of the commonest, and best understood types of synaesthesia. It is defined by the automatic experience of colour caused by perceiving letters, numbers, symbols, and words. The physicist Richard Feynman described his experience of it in his second autobiographical work, '*What do you care what other people think*?'; he wrote, "When I see equations, I see the letters in colors–I don't know why. As I'm talking, I see. . .light-tan j's, slightly violet-bluish n's, and dark brown x's flying around. And I wonder what the hell it must look like to the students" (p. 59).

There are a number of reasons grapheme-colour synaesthesia has been the focus of the most research. First, it has historical precedence as the first type to be well-documented [9–11]. The second reason relates to the relative ease with which it can be verified; the first and most well-developed screening tests for synaesthesia are for this form [6, 12, 13]. The third reason is that grapheme-colour synaesthesia is not all that rare. Estimates suggest that more than 1.4% of the general population have grapheme-colour synaesthesia; this means that worldwide there may be more than 90 million individuals who have it [13, 14]. Lastly, and perhaps the most important of these reasons is that grapheme-colour synaesthesia offers a unique lens through which to study and understand the biology of language [15–18].

Coincidentally, problems with specific parts of language such as reading, writing, and mathematics are thought to affect more than 5% of the general population; and there is no strong indication that this is different for synaesthetes [19–22]. Accordingly, we estimate that many millions of people have both grapheme-colour synaesthesia and such a problem with language. Despite sharing a common ground in language there is little known about how such problems might interact with synaesthesia. People at the intersection of these traits represent an under-researched and under-serviced population.

This research was motivated by a chance interaction with one such individual–SP–as we have reported in *Neurocase* [23]. Her plight culminated in the creation and testing of the Digit-Colour Calculator (DCC), an app that allows its users to select the colours with which it displays its digits. Quite unexpectedly, SP confirmed that the DCC was 'life-changing' after she put it to use for the very first time; this was, as SP explained, because the DCC made it "85% easier" to perform everyday calculation tasks that she had always experience difficulty carrying out.

The design of the DCC was based on an intuition that reinforcing the concurrent colours that SP experiences might reduce the cognitive load that she bears when performing calculations. We think that the DCC works by supplying some of the information that is represented by the colours SP experiences and, by doing so, reduces the need for SP to generate it solely by herself. Moreover, it seems plausible that by reducing the difference between the 'top-down' and 'bottom-up' signals in her brain–areas which are involved in processing such differences (e.g. subregions of the Anterior Cingulate Cortex and Thalamus) would face a lower load [24–27]. To the best of our knowledge the DCC is the first reported intervention that treats a concomitant disorder by reinforcing the phenomenology of a synaesthete.

In the study reported here, we sought to explore the potential benefits of the DCC in a broader sample of synaesthetes; i.e. in a group which had participants who did not present with the same struggle with calculations that SP faced. We wanted to know–would these individuals also show or perceive a benefit from the DCC?

We tested the performance of fifty-three grapheme-colour synaesthetes on a calculation and data-entry task similar to the one we used with SP. The test was designed to retain some ecological validity by mimicking the first use of the DCC–i.e., "organising. . . money"; in it the participant was required to complete a series of arithmetic calculations with the DCC, then enter their answers into a ledger. We also collected their qualitative responses to the DCC by means of semi-structured interview and questionnaire.

The effects of the calculator were contrasted across three conditions: *Control*, in which the colour-fills of digits were black; *Congruent*, in which the colour-fills were chosen to match the individual's associations; and *Incongruent*, in which the colour-fills were set to hues complementary to those from Congruent. In a second experiment we investigated the reliability of the recorded effects by re-testing the participants and contrasting their pattern of effects with those from a group of thirty-five non-synaesthetes. Finally, in a further experiment we identified the quartile of synaesthetes who had displayed the largest initial response (whether positive or negative) to the DCC and measured their response over a series of six tests to investigate the stability of these effects. We report the results and use them as a contextual frame in discussing the additional processing cost of synaesthesia.

## Methods

### Study advertising and sampling

The study was advertised to First Year Psychology students from The University of Sydney in accordance with their long-established research participation scheme; it was also advertised to the public via flyers posted on campus noticeboards, social media, and websites. Finally, word of mouth was used to encourage passive snowball recruitment. Those subjects not receiving course credit had their participation acknowledged for time and out-of-pocket costs with gift vouchers valued up to $50. The study was approved by the University of Sydney Human Research Ethics Committee and participants provided written consent prior to testing.

### Inclusion criteria and screening

We included participants over 16 years old who had normal or corrected-to-normal vision. Candidates who declared strong digit-colour associations next completed the online Synaesthesia Battery [12]. We used an inclusion criterion of a consistency score < 1.43, in keeping with previous research as it is shown to maximize sensitivity and specificity of the test [28]. We recruited the Non-Synaesthetes from First Year Psychology students.

### Experimental set-up

We conducted the study in a testing booth, under ambient light conditions, using a bespoke app running on a 12.9-inch Apple iPad Pro® [iOS 9.3.5]. The tablet was supported by a clamp and angled 30° to vertical and set 30 cm in from the edge of the table. We asked participants first to calibrate the calculator by pairing their own colours with the digits 0–9, using an embedded colour selection widget. For example, if a synaesthete experienced greenness when viewing the number 3, she would change its digit-fill colour to align with that green. The non-synaesthete participants were asked to arbitrarily assign a variety of colours that 'they like' to the digits, and these were used for their Congruent condition. To account for poor contrast between any *concurrents* with low saturation and the white background, the digits were bounded by a black border. Initially, the colour-digit assignments for the Incongruent condition were generated by rotating the hue metric of the selected colour 180˚ in the hue, saturation, value (HSV) colour space.

## Procedure

We familiarised participants with the layout and requirements of the task during a guided practice (Fig 1). They were instructed to complete the task as quickly and accurately as possible.

The main testing session consisted of three blocks with each comprising thirty task items (S1 Fig) and presented in one of the following conditions: *Control*, in which the colour-fills of digits were black; *Congruent*, in which the colour-fills were displayed to match the individual's specified associations; and *Incongruent*, in which the colour-fills were set to hues complementary to those from Congruent. For each participant the Congruent, Incongruent, and Control blocks were randomly assigned to an order of presentation and this Condition Order was counterbalanced over the group. The participants answered a questionnaire and participated in a semi-structured debrief interview (S2 Fig).

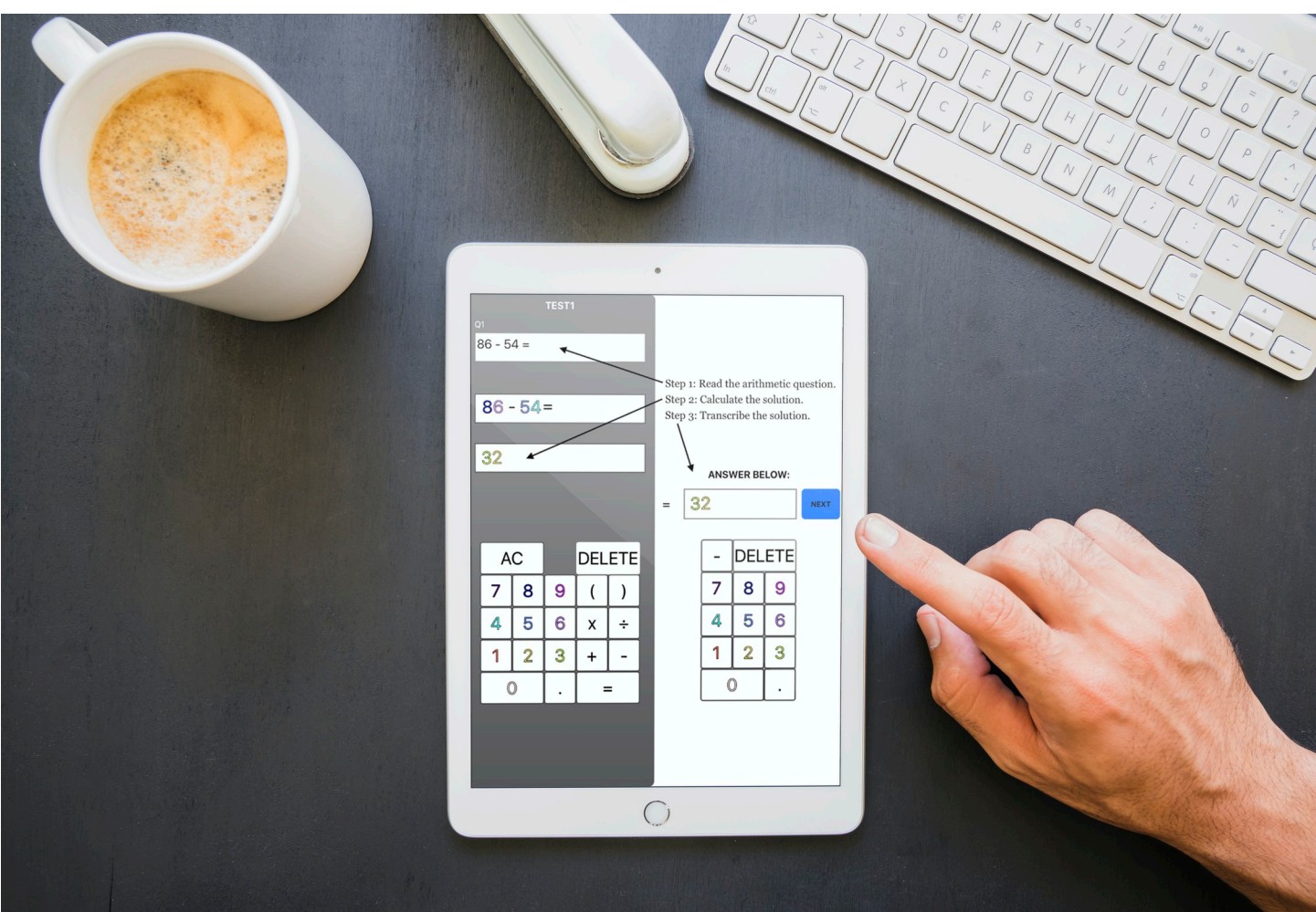

**Fig 1. A user interacts with the task interface as displayed on a tablet device.** The annotations indicate the three steps needed to complete each item which were (1) Reading: read an achromatically presented arithmetic problem that appeared in the upper left box; (2) Working: use the calculator number pad that is located at the lower left to solve the arithmetic problem; and (3) Transcribing; copy the solution across to the right via a congruously displayed answer pad, before pressing 'next'. The dependant time variable used in our analysis that was measured for each item was the response time for each item; it was measured from presentation of the item until the participant pressed 'next'. Note: this figure has been designed using content from Freepik.com and the annotations are for illustrative purposes only.

**A note on the analyses of results and accompanying issues relating to sample representativeness.** We used the statistical programming language R 4.0 and the BRMS (Bayesian Regression Models using Stan) package running within the R-studio 1.2 developer environment, to calculate Bayesian regression models from the study data [29, 30]. Detailed information on all the models can be found summarised in S2 Table. The BRMS package's default priors for the function-specific families were initially used for modelling. However, in modelling the second set of measures taken for the Synaesthetes we used the posterior distributions of the first model as experimentally informed priors. From these regression models we estimated population means along with their 95% credible intervals (CIs). The CIs are a region wherein an estimated parameter falls with some specified probability and can be interpreted as follows: "based on our model and from the results that were used to compute it we estimate there is a 95% probability that the population parameter exists within this region."

## Group experiment: Part 1

### Results

**Participants.** We initially recruited fifty-seven candidate grapheme-colour synaesthetes, thirty-two of whom responded to public advertising and twenty-five fulfilling First Year Psychology requirements. Four were excluded from analysis because their consistency scores on the online Synaesthesia Battery screening test were > 1.43. For further details on the sample, see Table 1.

**Quantitative results.** Accuracy rates were considered but as there was a ceiling effect imposed by the task design (in that there were few mistakes) no major conclusions could be drawn. There was no such ceiling effect on response times, so these were used to assess performance. The Synaesthetes' median response times ($\tilde{x}_{RT}$) by condition were 11.0 s (Congruent), 12.1 s (Incongruent), and 11.2 s (Control).

We computed a model to estimate the population's mean response time $\tilde{x}_{RT}$ (i.e. $\hat{\mu}_{RT}$) by condition. We modelled response time from the shifted lognormal family function using the formula Response Time ~ Condition + (1|ID).

The Synaesthetes' $\tilde{x}_{RTs}$ by condition are shown in Fig 2 –A as combined jitter and violin plots, with the regression model's estimates of $\hat{\mu}_{RT}$ superimposed within the 95% CI [Incongruent $\hat{\mu}_{RT}$ = 11.9 s, 95% CI (11.2 s, 12.7 s); Congruent $\hat{\mu}_{RT}$ = 11.0 s, 95% CI (10.5 s, 11.7 s); and

**Table 1. Group experiment: Part 1—The synaesthetes' (n = 53; 39 female and 14 male) descriptive statistics.**

| Category: | Age (Years) | # Types of synaesthesia declared | Consistency Score | VVIQ2 Score N = 52 | Projector/ Associator score | Education Level | Mathematics Level | Mathematics Ability | Mathematics Affinity |
|---|---|---|---|---|---|---|---|---|---|
| *mean* | 22 | 5 | 0.7 | 4 | -2 | N/A | N/A | N/A | N/A |
| *sd* | 9.2 | 3.9 | 0.23 | 0.7 | 1.4 | N/A | N/A | N/A | N/A |
| Med. | 21.5 | 5 | 0.6 | 3.9 | -1.7 | 2 | 1 | 4 | 4 |
| Q1 | 18.8 | 3 | 0.5 | 3.3 | -2.5 | 2 | 1 | 3 | 2 |
| Q3 | 27.7 | 8 | 0.8 | 4.4 | -0.7 | 3 | 2 | 5 | 6 |

A negative projector/associator score indicates an associator-type synaesthesia. Education Levels were scored thus: High School incomplete = 0, High School or equivalent diploma complete = 1, Enrolled in a Bachelor Level course = 2, Completed a Bachelor Level course = 3, Enrolled in a postgraduate degree = 4, Completed a postgraduate degree = 5. Mathematics Levels were scored thus ~ No mathematics for the final years of school = 0, Basic mathematics for the final year of school = 1, Bachelor level mathematics or advanced school mathematics = 2, Advanced university mathematics = 3. Mathematics Ability and Mathematics Affinity were self-reported on Likert scales (1 = Extremely Weak to 7 = Extremely Strong) and (1 = I hate it to 7 = I love it), respectively. Med. = median; Q1 = the first quartile; Q3 = the third quartile; and N/A = Not Applicable.

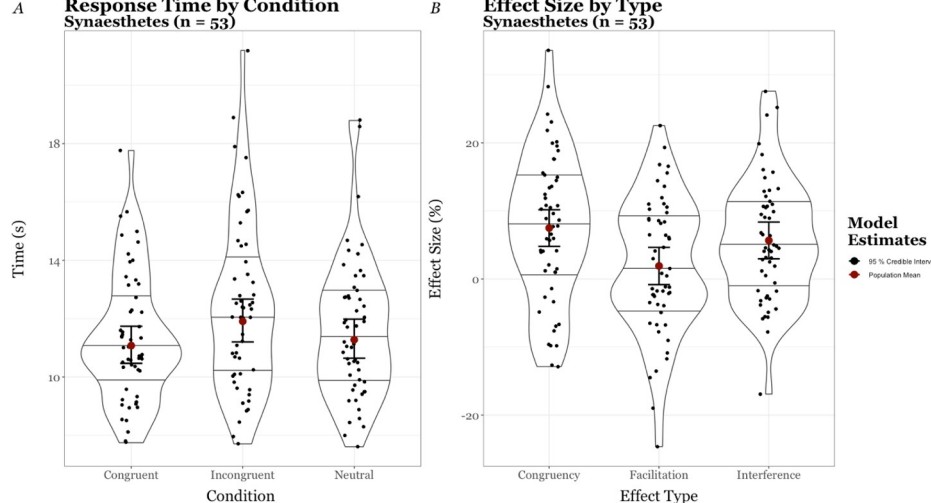

**Fig 2. Synaesthetes' response times.** Panel A–Violin plots with superimposed jitter plots for synaesthetes' (n = 53) median item times (s) by condition with accompanying model estimates of the population mean and Panel B–Violin plots with superimposed jitter plots for synaesthetes' (n = 53) effect size by type with accompanying model estimates of the population effect sizes.

Control $\hat{\mu}_{RT}$ = 11.3 s, 95% CI (10.6 s, 12.0 s]. Additionally, we calculated three standardised effect sizes from each participant's $\tilde{x}_{RT}$, as follows:

$$\text{the Interference Effect} = \frac{Incongruent_{\tilde{x}_{RT}} - Control_{\tilde{x}_{RT}}}{(Incongruent_{\tilde{x}_{RT}} + Control_{\tilde{x}_{RT}})/2} \times 100; \tag{1}$$

$$\text{the Congruency Effect} = \frac{Incongruent_{\tilde{x}_{RT}} - Congruent_{\tilde{x}_{RT}}}{(Incongruent_{\tilde{x}_{RT}} + Congruent_{\tilde{x}_{RT}})/2} \times 100; \tag{2}$$

$$\text{the Facilitation Effect} = \frac{Control_{\tilde{x}_{RT}} - Congruent_{\tilde{x}_{RT}}}{(Congruent_{\tilde{x}_{RT}} + Control_{\tilde{x}_{RT}})/2} \times 100. \tag{3}$$

We computed a second model with the formula Effect Size ~ Effect Type + (1|ID) from the skew normal family function to estimate the $\hat{\mu}$ size of these effects; these are superimposed over the sample's distributions, in Fig 2B. Our model's estimates [$\hat{\mu}_{congruency\ Effect}$ = 7.5%, 95% CI (4.8%, 10.2%), $\hat{\mu}_{Interference\ Effect}$ = 5.7%, 95% CI (2.9%, 8.5%), and $\hat{\mu}_{Facilitation\ Effect}$ = 1.9%, 95% CI (-0.8%, 4.7%)] are strong evidence for both a population level *Congruency* and *Interference Effect*; but weak evidence that a marginal *Facilitation Effect* exists.

**Questionnaire results.** A majority of synaesthetes (> 75%) ranked the Congruent condition easiest ($\chi^2$ (2, 53) = 64.99, p < 0.001 (See Table 2A)), while only one participant found it the hardest. Participants variously reported: visual search facilitation, "I could just look for the colour and I could sort of see the colour in the corner of my eye without having to actually focus on the shape of the number"; an increased sense of cognitive ease, "The calculator with my colours, I found to be quite fluid, like the easiest one"; and increased determination, "I was happy when my colours were there and determined to do the things quicker". The 20% who found Control easiest remarked on its relative familiarity compared with the novelty of using a colourfully presented DCC. For example, "it's weird to see them, in my colours . . . with the

**Table 2. Participants' (n = 53) questionnaire results as derived from their answers to the questionnaire.** (a) Difficulty Rankings. (b) Likert Rankings.

**(a)**

| Condition | Easiest | Hardest |
|---|---|---|
| Congruent | 40 | 1 |
| Control | 9.5 | 17 |
| Incongruent | 3.5 | 35 |

**(b)**

| Ranking | 1 | 2 | 3 | 4 | 5 | 6 | 7 |
|---|---|---|---|---|---|---|---|
| *Count* | 0 | 1 | 3 | 5 | 11 | 10 | 23 |

(a) Difficulty rankings of the tests by the condition. Note: scores of 0.5 were assigned to participants who ranked conditions equivalent. (b) Counts of the Likert rankings (1 = No, Never–through 4 = even-chance/ possibly–to 7 = Yes, Definitely) to the question: "If possible, would you use a calculator that showed your colours in the future?".

black one I was like, 'I have done this before'". The one participant who found Congruent to be the hardest commented, "It was a disconcerting experience. . . the colours are very personal, it felt like my privacy had been invaded". When ranking the likelihood of using a DCC in the future, on a 7-point Likert scale (See Table 2**B**), > 40% of synaesthetes indicated that they would certainly use a DCC in the future, if one were available. A Wilcoxon Signed Ranks Test performed on the data with a $H_{null}$: $\tilde{x}$ (Sample Median) = 4 –i.e. the sample being equivocal in their indications of the use of a DCC in the future–indicates strong evidence (V = 1133.5, p < 0.001) against $H_{null}$ and estimates $5.5 < \tilde{x} = 6 < 6.5$, 99% confidence interval.

## Discussion

The quantitative results corroborated those that are generally reported in the Synaesthetic-Stroop literature: that is, a group-level performance cost for the synaesthetes when completing a task in an incongruent condition compared with either an achromatic control or congruent condition (Interference and Congruency Effects, respectively), and only a marginal difference in performance between Congruent and Control (Facilitation Effect). The participants' verbal and written responses to the DCC which displayed their colours were overwhelmingly positive.

Despite synaesthetes being known to experience increased Emotionality, they have been found not to identify, analyze, or verbalise their emotion in an increase manner [31]. In line with this, we suspect the emotive language used by the synaesthetes was a genuine response to the personal nature of the intervention rather than some general tendency to emotive language. We also note that blinding was limited because of the experimental design, but because of the overwhelming strength of these reports we expect the DCC and similar tools that reinforce synaesthesia will be pleasing to many.

The collected measures which described the demographics, the individuals' synaesthesia and the relationship each participant had with mathematics failed to explain what might distinguish between the individuals who showed the most benefit from the DCC and those who did not. Despite this, we were encouraged by the group's preference for the Congruent DCC, over the standard achromatic Control, and by finding that some individuals may be faster using it. On the basis of the results, which indicate that the DCC could be a desirable and valuable tool, we decided to examine its inter-test reliability, as well as establish its sensitivity to synaesthesia, by testing a non-synaesthete group.

## Group experiment: Part 2

### Method

Feedback from part 1 indicated that for dark colours, the Incongruent and Congruent stimuli were difficult to distinguish on occasion. For example, one participant indicated that '8' displayed in a very dark green, as presented during Incongruent, 'felt' similar to her '8' as presented with a very dark purple during Congruent. Accordingly, the process to assign colours for Incongruent was modified in part 2 to provide also the complementary shade of the Congruent colour. In this instance this had the effect of changing a dark green '8' to a bright purple '8'. This was achieved by setting the $Value_{Incongruent} = 1 - Value_{Congruent}$ in the HSV colour space. Two new items were added to the questionnaire in order to examine the synaesthetes' beliefs about the use of other tools, such as a colourful keyboard, which might substantiate their *concurrent*s for letters as well as digits. We also recruited participants for a qualitative control group, the Non-Synaesthetes, and all were randomised to a new Condition Order which was then balanced at the group level. See Fig 3 for a demonstrative contrast of the colours assigned by the Synaesthetes and Non-Synaesthetes to their digit (0–9) stimuli.

### Results

**Participants.**  forty-three of the original fifty-three synaesthetes returned for a second session. The Non-Synaesthetes (n = 35) were a new group of First Year Psychology students who participated for course credit and who declared they had no strong grapheme-colour associations. For further details on the sample, see Table 3.

**Colour assigned by the participants for the digit stimuli (0-9)**

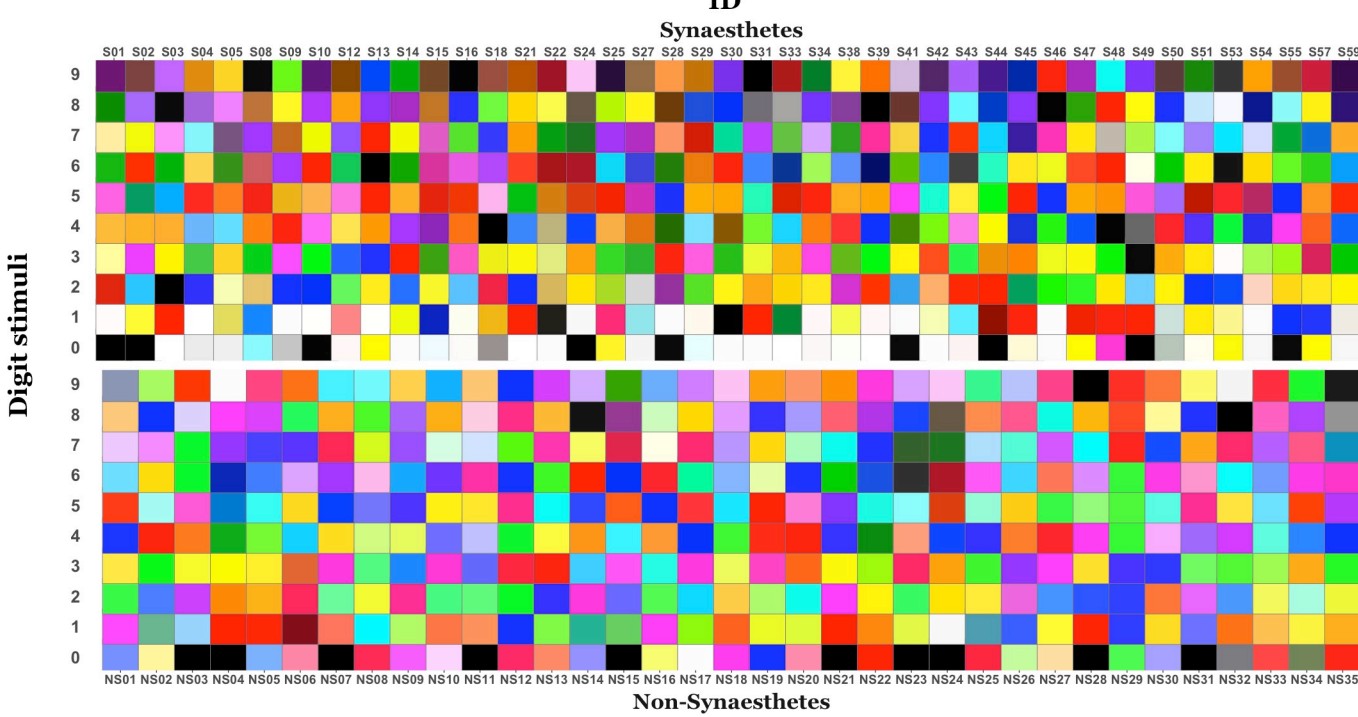

**Fig 3. Colour swatches demonstrating the different colours assigned by each participant for the digits of their congruent condition: Synaesthetes (n = 43) | non-synaesthetes (n = 35).**

**Table 3. Group experiment: part 2—synaesthetes (n = 43; 32 female and 11 male) and non-synaesthetes (n = 35; 32 female and 3 male) descriptive statistics.**

| Category: | | Age (Years) | Education Level | Mathematics Level | Mathematics Ability | Mathematics Affinity |
|---|---|---|---|---|---|---|
| Synaesthetes | *mean* | 25.77 | N/A | N/A | N/A | N/A |
| | *sd* | 9.9 | N/A | N/A | N/A | N/A |
| | Med. | 22 | 2 | 1 | 4 | 4 |
| | Q1 | 19 | 2 | 1 | 3 | 2 |
| | Q3 | 26 | 3 | 2 | 5 | 6 |
| Non-Synaesthetes | *mean* | 19.35 | N/A | N/A | N/A | N/A |
| | *sd* | 0.9 | N/A | N/A | N/A | N/A |
| | Med. | 19 | 1 | 2 | 5 | 5 |
| | Q1 | 19 | 1 | 1 | 4 | 4 |
| | Q3 | 20 | 1 | 2 | 6 | 6 |

**Quantitative results.** The Synaesthetes and Non-Synaesthetes $\tilde{x}_{RTs}$ were 10.9 s and 9.9 s, respectively; between conditions the $\tilde{x}_{RTs}$ were 10.6 s (Congruent), 11.1 s (Incongruent), and 10.8 s (Control) for the Synaesthetes and 10.1 s (Congruent), 9.9 s (Incongruent), and 9.7 s (Control) for the Non-Synaesthetes. We computed two models to estimate each population's mean $\tilde{x}_{RT}$ (i.e. $\hat{\mu}_{RT}$) by condition. For each, we modelled response times from the shifted log-normal family function using the participants' results and the formula Response Time ~ Condition + Condition + (1|ID). We used the estimates from the posterior distribution of the first model as priors when modelling for the Synaesthetes and did not specify informative priors for the Non-Synaesthetes. We then plotted the models' estimates for the $\hat{\mu}_{RT}$s, within their 95% CI for the Non-synaesthetes [Congruent $\hat{\mu}_{IT}$ = 10.1 s, 95% CI (9.5 s, 10.8 s); Incongruent $\hat{\mu}_{IT}$ = 10.1 s, 95% CI (9.5 s, 10.8 s); Control $\hat{\mu}_{IT}$ = 10.0 s, 95% CI (9.4 s, 10.6 s)] and for Synaesthetes [Congruent $\hat{\mu}_{IT}$ = 10.7 s, 95% CI (10.2 s, 11.2 s); Incongruent $\hat{\mu}_{IT}$ = 11.3 s, 95% CI (10.7 s, 12.0 s); Control $\hat{\mu}_{IT}$ = 10.8 s, 95% CI (10.3 s, 11.4 s] over the distribution of each group's $\tilde{x}_{RTs}$ by condition (see Fig 4A).

As was done in part 1, we also calculated three standardised effect sizes from each participant's $\tilde{x}_{IT}$ and modeled from them; these are displayed in Fig 4B, along with the population estimates. For the Non-Synaesthetes the model's estimates [$\hat{\mu}_{Congruency\ Effect}$ = -0.3%, 95% CI (-3.1%, 2.5%); $\hat{\mu}_{Interference\ Effect}$ = 1.0%, 95% CI (-1.8%, 3.9%); and $\hat{\mu}_{Facilitation\ Effect}$ = -1.4%, 95% CI (-4.2%, 1.4%)] suggest only limited evidence for the potential of population-level effects. For the Synaesthetes our model's estimates [$\hat{\mu}_{Congruency\ Effect}$ = 7.0%, 95% CI (5.0%, 9.1%); $\hat{\mu}_{Interference\ Effect}$ = 5.7%, 95% CI (3.4%, 7.8%); and $\hat{\mu}_{Facilitation\ Effect}$ = 1.7%, 95% CI (-0.5%, 4.0%)] corroborate well–as might be expected with our informative priors–those from the model calculated in part 1, i.e. suggesting overall the results do provide strong evidence that Synaesthetes would display a *Congruency Effect* and an *Interference Effect* at the population level; however the results provide only moderate evidence they would display a *Facilitation Effect* at the population level.

**Questionnaire results.** A significant majority of Synaesthetes (> 85%) ranked the Congruent condition easiest [$\chi^2$ (2, 43) = 68.946, p < 0.001] and none ranked it hardest (See Table 4a). The Synaesthetes made a range of comments such as, "During [Control] I felt more tired and that it was more difficult to correctly order the numbers" and "I feel like my colours helped me with like memory". The Non-Synaesthetes were equivocal in their rankings, with a Chi-Squared test providing strong evidence for this [$\chi^2$ (2, 35) = 4.7155, p = 0.95], and in their reporting of the perceived difficulty of the test between conditions. When the Synaesthetes were questioned on whether they would use a DCC in the future; whether they believed

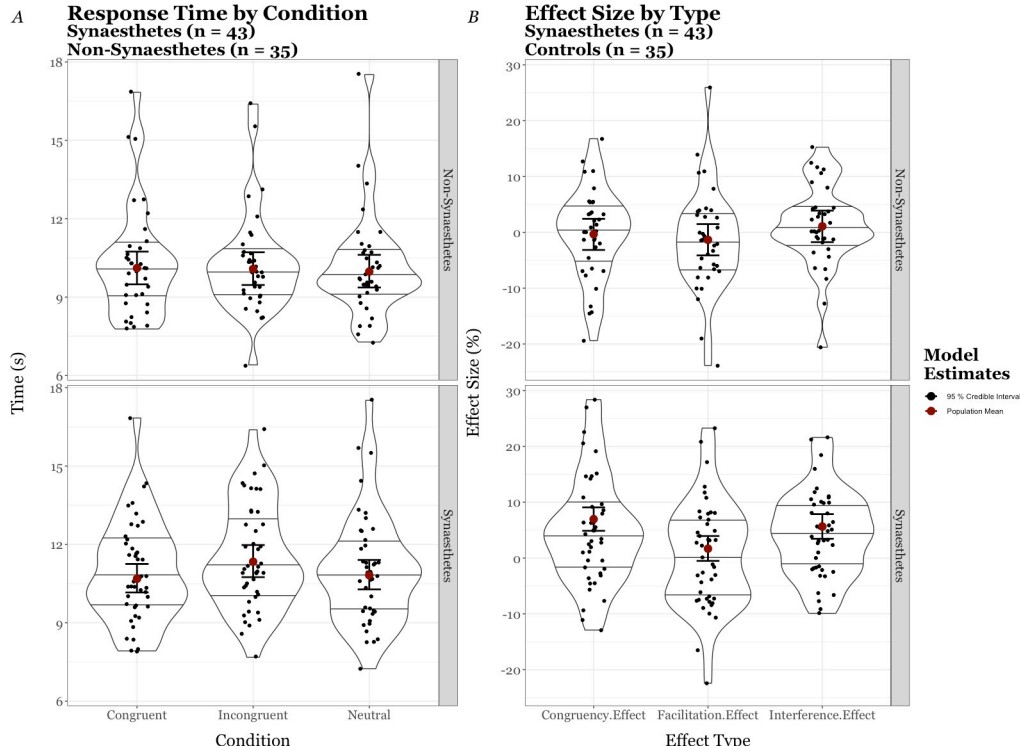

**Fig 4.** Panel A–Violin plots with superimposed jitter plots of Synaesthetes' (n = 43) vs Non-Synaesthetes' (n = 35) median item times(s) by condition with accompanying model estimates of the population mean, and Panel B–Violin plots with superimposed jitter plots of Synaesthetes' (n = 53) vs Non-Synaesthetes' (n = 35) Effect size by type with accompanying model estimates of the population effect size.

analogous tools might be useful; and whether such tools may have been helpful earlier in their lives, they responded very positively (Table 4**b**). Wilcoxon Signed Ranks Tests performed on the Likert data with the common $H_{null}$: $\tilde{x} = 4$ indicates strong evidence against each [Question

**Table 4. Synaesthetes' (n = 43) and non-synaesthetes (n = 35) questionnaire results as derived from their answers to the questionnaire.** (a) Difficulty Rankings. (b) Likert Rankings.

| (a) | | | | | | |
|---|---|---|---|---|---|---|
| **Condition** | | | Easiest | | Hardest | |
| | | | *Synaesthetes* | *Non-Synaesthetes* | *Synaesthetes* | *Non-Synaesthetes* |
| Congruent | | | 35 | 12 | 0 | 9 |
| Control | | | 5.5 | 14 | 12 | 9 |
| Incongruent | | | 0 | 6 | 31 | 14 |
| (b) | | | | | | |
| **Ranking** | 1 | 2 | 3 | 4 | 5 | 6 | 7 |
| Q1 Count | 0 | 1 | 1 | 2 | 9 | 7 | 23 |
| Q2 Count | 0 | 0 | 1 | 1 | 11 | 6 | 24 |
| Q3 Count | 0 | 1 | 3 | 6 | 7 | 9 | 17 |

**(a)** Difficulty rankings of the tests by the condition. **(b)** Counts of the Synaesthetes (n = 43) Likert rankings (1 = No, Never–through 4 = even-chance/ possibly–to 7 = Yes, Definitely) to the following questions: Q1, "If possible, would you use a calulator that showed your colours in the future?"; Q2, "Do you think other devices, that work by the same principle i.e. showing your synaesthetic colours, would be useful?"; and Q3, "Do you think, growing up, having a tool like this may have helped?".

1 (V = 841, p < 0.001); Question 2 (V = 896.5, p < 0.001); and Question 3 (V = 671, p < 0.001)] with sample estimates of $6 < \tilde{x} = 6.5 < 7$, 99% CI; $5.5 < \tilde{x} = 6 < 7$, 99% CI; and $5.5 < \tilde{x} = 6 < 6.5$, 99% CI, respectively. These rankings reflect comments made by the Synaesthetes e.g. '[Control] was easier than [Congruent] but it did feel more homely and this might have made maths less boring growing up'; 'I did well [growing up] without it but, it would have been nice'; 'Colourful keyboards would be awesome'; and 'I would have absolutely loved a calculator like this as a kid. I loved English, Languages, History, Art but Maths was always confusing and I truly believe this would have helped me'.

## Discussion

The results showed that the Synaesthetes were around 10% slower than the Non-Synaesthetes to complete the timed calculator task. However, underlying sample differences in mean age and levels of education limit the insights drawn from a direct comparison of these groups (S1 Table). The effects seen for the Synaesthetes in part 1 were replicated, albeit with smaller magnitudes, demonstrating a reasonable inter-test reliability. In contrast, and in general agreement with the Synaesthetic-Stroop literature, the Non-synaesthete group showed no such effects. We also note that questionnaire results similarly contrast the Synaesthete and Non-Synaesthete groups: the Synaesthete group reported that they found Congruent easiest and liked the notion of tools that substantiated their synaesthesia, whereas the Non-synaesthete group were equivocal when comparing conditions. Importantly, the synaesthete who was initially perturbed by viewing his own colours 'out there' had overcome his surprise and now reported Congruent to be the easiest.

The results of this group experiment suggest there are at least subjective benefits for Synaesthetes from the DCC that substantiates their *concurrent* over ones which were Incongruent or Control. To facilitate further analysis, we undertook additional testing on a subsample of the Synaesthetes from part 2.

## Subsample experiment: Repeated measures

### Aim

In this experiment, we wanted to examine whether the synaesthetes who had demonstrated an extreme effect (in the top quartile) on speed from the Congruent calculator would continue to show such a response, or whether these effects might merely be driven by novelty, and thus dissipate over repeated testing.

### Method

To select the subsample of extreme responders we employed two different methods. In the first, we averaged each synaesthete's (n = 43) *Facilitation Effect* from parts A and B of the Group Experiment. We then identified the top quartile of participants by the magnitude of their *Facilitation Effect*$_{\text{UnadjustedAverage}}$. Because Condition Order was only balanced at the group level for parts A and B, we also employed a second selection method to ensure our selection identified all the extreme responders. In this second method, we individually adjusted each participant's *Facilitation Effect* size by the difference between her or his Condition Order's mean *Facilitation Effect* size and the mean *Facilitation Effect* size from the overall group, before averaging the result over part 1 and 2 (Table 5 and S3 Fig).

Over a 12-month period all these individuals were invited back for further study and to complete the same task as before. For this experiment, each participant completed the task six times in order to balance the Condition Order on the individual level.

**Table 5. The subject IDs and effect sizes for the top quartile of participants as selected by their adjusted and unadjusted *facilitation effects_{Average}* from the group experiment.**

|  | (-) Effect, n = 2 | | (+) Effect (n = 9) | | | | | | | | |
|---|---|---|---|---|---|---|---|---|---|---|---|
| *Subject ID* | SYN059 | SYN009 | SYN016 | SYN053 | *SYN031* | SYN043 | SYN057 | SYN004 | SYN025 | SYN055 | SYN024 |
| *Effect Size (%) Adjusted* | 11.31 | 10.36 | 7.75 | 8.00 | 8.04 | 8.42 | 8.99 | 11.11 | 12.58 | 12.83 | 14.32 |
| *Subject ID* | SYN009 | SYN059 | SYN055 | SYN025 | SYN057 | *SYN045* | SYN053 | SYN043 | SYN024 | SYN016 | SYN004 |
| *Effect Size (%) Unadjusted* | 17.74 | 14.46 | 8.58 | 10.81 | 11.31 | 11.45 | 12.42 | 12.72 | 13.23 | 13.83 | 14.55 |

## Results

Both methods used to identify extreme responders yielded near identical groups: each method identified only one single participant (SYN045 & SYN031, respectively) that the other did not. Both selection methods produced two subgroups. Subgroup A consisted of ten participants who were faster during the congruent condition over the Group Experiment. These participants all rated Congruent easiest and all ranked 7/7 to the question probing future use of a DCC. Subgroup B consisted of a pair of participants (SYN009 & SYN059) who had been slower using the DCC during Congruent. Their slower speeds stood in apparent contrast to their reports: they both found Congruent easiest and also ranked 7/7 on a Likert scale for the question probing their potential future use of a DCC. *Participants*. Of the twelve participants identified as eligible, nine were available for follow-up study: seven of the ten from the faster Subgroup A (6 females and 1 male, $Age_{mean}$ = 29.7 years, sd = 9.6) and both from the slower Subgroup B (both males, aged 22.3 and 32.8 years).

**Quantitative results.** The participants from the Subgroup A continued to demonstrate a speed benefit for thirty-six of the following forty-two tests (See Fig 5; Row A–Facilitation Effect); this meant they displayed a speed benefit over both experiments for fifty out of the

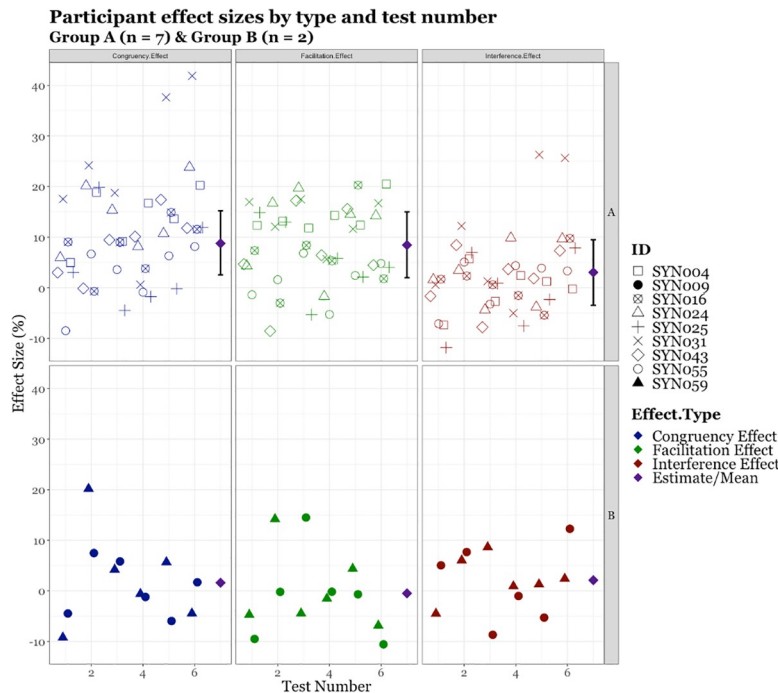

**Fig 5. Participant effect sizes by test, type, and group with model estimates for subgroup A and B.**

fifty-six tests. Two participants (SYN004 and SYN031) demonstrated a benefit eight out of the eight times they were tested. A multi-level model was calculated for the size of Subgroup A's effects. This model suggests good evidence that the underpinning population of synaesthetes who initially show a marked response to the DCC would continue to demonstrate a robust *Facilitation Effect* [$\hat{\mu}$ = 8.4%, 95% CI (1.97%, 14.99%)] and a *Congruency Effect* [$\hat{\mu}$ = 8.8%, 95% CI (2.54%, 15.20%)] but no strong evidence for an *Interference Effect* [$\hat{\mu}$ = 3.0%, 95% CI (-3.48%, 9.48%)]. No estimates were calculated for the two subjects in Subgroup B as the sample size was too small to allow for appropriate modelling. However, when examining their results (as displayed in Fig 5) it is clear they did not show any consistent effects over the repeated testing (See Fig 5; Row B)

## Discussion

The participants from Subgroup A demonstrated a robust speed advantage from using a DCC that displayed their concurrent colours. Moreover, their self-reports suggest that this benefit was strongly perceived. Unlike the Synaesthete group at large however, there was no strong evidence to suggest Subgroup A were meaningfully worse when completing the task in Incongruent compared with Control colours. Indeed, the participants' subjective reports corroborated this: five of the seven participants rated the Incongruent block easier than Control. These five showed a general preference for coloured stimuli over achromatic ones and made telling comments like, '*When it was the other colours and not mine, it was kind of like someone who wants to make friends with you, but they don't really get you. I found the black . . . very impersonal*'. This interesting result suggests that perhaps, for some grapheme-colour synaesthetes, the normally encountered black graphemes of Control are closer to Incongruent than Congruent. It may also explain why some synaesthetes report such difficulty with ordinary math tasks. Our prediction that measures of low affinity and ability for mathematics might best predict a benefit of the DCC proved incorrect: Indeed, it was SYN004 and SYN031 who showed the most robust benefits and they were two of our faster ($\mu_{\bar{x}_{IT}}$ = 8.4 s and 10.2 s, respectively) participants that gave high ratings for mathematical affinity (7/7 and 5/7) and ability (7/7 and 5/7), respectively. Moreover, even our fastest participant SYN025 ($\mu_{\bar{x}_{IT}}$ = 7.9 s), who rated 6/7 and 7/7 for mathematical affinity and ability respectively, demonstrated a benefit for 7 of the 8 times she was tested. This is interesting because while it seems plausible that motivation effects could explain a benefit to participants in the middle and bottom ranges of performance, it seems less likely that such effects would explain the benefits to those already performing at the top. The results of Experiment 2 also indicate that when repeatedly tested, no participants showed a consistent cost to performance. We suspect the initial detriment to speed (i.e. in the Group Experiment) for the pair from Subgroup B was an effect from the novelty, rather than reflecting any perceptual effects.

## General discussion

This study follows our single case report on SP [23], in which we described the testing of a novel tool, the Digit-Colour Calculator (DCC); this tool helped SP overcome the struggle with calculations that she faced, for example when she was "organising money". In her words the effect was "life changing". For the current study we have explored whether this tool might be of use for other synaesthetes who did not face the same struggle.

Our results agree with those reported in the literature, which show that people with grapheme-colour synaesthesia perform better on a task when it uses graphemes displayed in the colours of their concurrents, in contrast to performance if they are displayed in different colours; this is known as the Congruency Effect [4, 32–35]. At the group level the Synaesthetes showed

a clear cost to speed when dealing with an incongruently coloured DCC. Moreover, they reported a very strong preference for undertaking the task with a DCC that displayed their colours. In contrast, the Non-Synaesthetes did not display such effects or report a strong preference.

The results of this study extend other commonly reported findings. Reporting of a comparison between the results of synaesthetes on a task when it uses graphemes displayed in black versus colour is less common. Perhaps this is because synaesthetes sometimes have black as their concurrent, which means as a condition it can vary between Incongruent and Congruent [36]. Moreover, because it is common, essentially normal, to see writing in black script the effects are confounded with those from exposure. For this very reason, however, it seems one the most practical comparisons to make. The results from our comparison with a standard calculator therefore provide an important insight. The group of synaesthetes overwhelmingly reported preferring the DCC that displayed their colours over the standard control. Their results and our parameter estimations also suggest the possibility of marginal performance benefits, beyond those subjectively felt, on a population level. Of course, our parameter estimates are affected by the same biases, relative to the actual populations, that our samples display–for example they would be skewed by the effect of the preponderance of females and high level of education within each group. We suspect that the latter bias will have reduced the magnitude of any estimated effect sizes compared with those that might be observed in the general population, because of its correlation with many favourable outcomes. Our general conclusions based on the results and their accompanying estimates are, therefore, made in this context. Indeed, it is probable that the bias underpinning our sample of synaesthetes (e.g. young age, a high level of education, a high affinity for mathematics) means that our estimates for the size of such effects are conservative. These performance effects are small, so any real-world benefit for the wider synaesthete population would most probably be limited to their subjective experience. Perhaps most crucially, however, the results of this study also show that even some synaesthetes who do not have difficulty in performing calculations also gain a genuine performance benefit from the DCC.

The task that we designed to test the effects of the DCC reflected its first use, 'organising money'. That is, the participants performed a series of calculations while entering them into a ledger analog. This meant that the behaviour retained some ecological validity, but there was a trade off with an increased complexity. Because of this there are many possible mechanisms that could explain the individual benefits that were observed. Indeed, the comments made by the synaesthetes from this study suggest that the DCC may have improved their motivation, and/or reduced the cognitive load of the task, and/or aided in the visual search component. Each example of a benefit may be best explained by some unique combination thereof.

This notion is further supported by the following two observations. First, the synaesthete population is diverse. Synaesthetes vary widely in almost all measures. For example, the reports on their phenomenology; their brain activity; their genetics; and demographics [37–39]. Secondly, there was no clear characteristic of subgroup A that simply predicted their facilitation— we had to select this group from their initial performance. [It is worth noting that the post-hoc selection by performance for *further testing* when no obvious predictors exist or have been measured (as might be expected to happen when testing in a small heterogeneous sample) is a valid method distinct from, but easily confused with, cherry picking [40]. For example, subgroup A was not simply those who disliked mathematics or felt they were particularly poor with calculation. Indeed, it was one of the speediest subjects (SYN004), highly educated and declaring the maximum (7/7) for her affinity and ability for mathematics on Likert scales, who showed the most consistent benefit.

Regardless, the aim in this study was not to determine the relative contributions of such mechanisms, but to establish if the DCC had application beyond its initial therapeutic use in our index subject. The results demonstrate that it does. In the next section, we propose a framework for understanding how the DCC and similar tools could work.

## An additional processing cost of synaesthesia

Synaesthesia exists widely as a trait in the general population and it is probable that at least 4% of us have it [14]. As with other traits, synaesthesia can have positive or negative effects. For example, some people with synaesthesia display prodigious memory as Luria noted in *The mind of a mnemonist*. Others like Julie Roxburgh, who was studied in detail by Baron-Cohen et al. in 1996 and featured in the BCC documentary *Orange Sherbet Kisses*, feel as if "...every one of [their] senses is being battered". Positive and negative effects of synaesthesia are also recorded at the population level [22, 41, 42]. In most cases, synaesthesia may simply exist as part of naturally occurring neurodiversity.

Regardless of the end effects, we are of the view that each synaesthetic experience bears an additional cost in the near term when compared with its equivalent example from typical perception. For example, the experience of music-colour synaesthesia *prima facie* bears some additional cost when compared with that paid for the mere 'ordinary' experience of hearing the music. This assertion follows from two premises. First, conscious perception is based on information processing in the brain and this has a cost that can be framed in terms of energy. Second, synaesthesia is an unusual type of perception which by definition has more components than normal conscious experience; namely, the concurrents.

If these premises are valid, there must be *at least* some marginal processing cost to create and sustain the perceptual divergence. We do not suggest that synaesthesia is costly, *per se*. Indeed, as already noted, this initial cost can pay dividends in terms of memory and increased perceptual acuity by way of enhanced processing [43, 44]. Moreover, this additional processing cost can be minimized at higher levels of cognition. For example, people with synaesthesia generally do not hold beliefs that their *concurrent(s)* are experienced by others; rather they often describe their *concurrent(s)* as lacking *perceptual presence* [8, 45]. Irrespective of any potential benefits or cost minimizations, one defining characteristic of synaesthesia can be seen as this additional processing.

## Reducing the cost

Within an ecological framework, the cost can be minimized by either *normalising* the perception of the synaesthete, or by aligning the environment to match the concurrent perception [46]. The first option seems undesirable and/or unattainable in most cases, given that trait synaesthesia is not commonly considered a pathology and the majority of people with synaesthesia consider it an inextricable part of their identity. Alternatively—as was the motivation for the DCC–the external stimulus can be brought into alignment with an individual's *concurrent* perceptions. Importantly, by engaging rather than eliminating an individual's synaesthesia, this strategy should retain any advantages the synaesthesia may confer. Moreover, it could be strategically leveraged (e.g. searching for colour rather than form in the case of the DCC). With the advent of personalised therapies, the ubiquity of smart technologies, and the further social acceptance of neurodiversity, this option may be regarded as both viable and attractive, and may serve the further investigation of such processing costs.

## Significance and limitations of the study

The results from this study establish that the DCC can provide a wider benefit to grapheme-colour synaesthetes beyond its therapeutic application for our initial case SP. Within our

sample of synaesthetes the DCC demonstrably improved speed for more than 13% and was strongly liked by more than 75%; importantly, the DCC did not hinder the performance of any of the synaesthetes. More generally, this study helps establish proof of concept that substantiating *the concurrent* is a viable and attractive principle in creating tools for synaesthetes. The prevalence of dyscalculia and dyslexia in the general population is around 4% and there is no strong reason to suspect a great a difference for the synaesthete population [19, 22]. Therefore, we believe there are millions with concomitant disorder who would stand to benefit from such tools. Moreover, we envisage such approaches could also prove 'life-changing' to others if they help overcome problems during foundational periods of learning.

The study faces a number of limitations inherent in exploratory work and common amongst synaesthesia research [47]; these relate to blinding, sampling, and the identification of appropriate control participants. While it has been shown possible to conduct well-blinded experiments with synaesthetes [48], the nature of the current experimental tasks made this impracticable. Regarding control participants, in our study the Non-Synaesthete group used in 'Group Experiment: part 2' were not perfectly matched on demographic measures with the Synaesthete group, which may reduce the validity of the comparison with this group. Additionally, there was no comparison group used in the 'repeated measures' experiment, where the selected group of synaesthetes completed the full order replication of the task conditions. However, this limitation does not significantly affect our conclusions, given that the design of this study was one of within-participant control where the critical comparisons of interest are between different conditions administered to the same participant. The Non-Synaesthete group tested in 'part 2' nevertheless provides a qualitative comparison with the Synaesthete group, showing that Non-Synaethetes are not significantly affected by variations in the colour displays in either their performance or their subjective preference.

## Conclusion

The results of this study show that the DCC can be useful beyond its original therapeutic application for the single subject, SP. We believe there is scope for further applied research to create analogous tools to reinforce synaesthetes' concurrents, and which can serve to explore the additional processing cost inherent to synaesthesia.

## Supporting information

**S1 Fig. Task items.**
(PDF)

**S2 Fig. Debrief questionnaires.**
(PDF)

**S3 Fig. Average facilitation effect sizes from group experiment part 1 and part 2.**
(PDF)

**S1 Table. Sample characteristics from group experiment part 2.**
(PDF)

**S2 Table. Parameter and effects information for the regression modelling.**
(PDF)

## Acknowledgments

We thank the Sydney Informatics Hub, a Core Research Facility of the University of Sydney, for their technical assistance. Finally, we thank all the participants of this study for their time and contributions.

## Author Contributions

**Conceptualization:** Joshua J. Berger, Irina M. Harris, Karen M. Whittingham, Zoe Terpening, John D. G. Watson.

**Data curation:** Joshua J. Berger.

**Formal analysis:** Joshua J. Berger, Irina M. Harris.

**Funding acquisition:** Joshua J. Berger, John D. G. Watson.

**Investigation:** Joshua J. Berger.

**Methodology:** Joshua J. Berger, Irina M. Harris, Karen M. Whittingham, Zoe Terpening, John D. G. Watson.

**Project administration:** Joshua J. Berger.

**Resources:** Joshua J. Berger, John D. G. Watson.

**Software:** Joshua J. Berger.

**Supervision:** Irina M. Harris, Karen M. Whittingham, Zoe Terpening, John D. G. Watson.

**Validation:** Joshua J. Berger.

**Visualization:** Joshua J. Berger.

**Writing – original draft:** Joshua J. Berger.

**Writing – review & editing:** Joshua J. Berger, Irina M. Harris, Karen M. Whittingham, Zoe Terpening, John D. G. Watson.

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
