## [Decision Letter · Decision Letter 0]

8 Feb 2021

PONE-D-20-35946

Sharing the load: How a personally coloured calculator for grapheme-colour synaesthetes reduces the additional processing costs.

PLOS ONE

Dear Dr. Berger,

Thank you for submitting your manuscript to PLOS ONE. After careful consideration, we feel that it has merit but does not fully meet PLOS ONE’s publication criteria as it currently stands. Therefore, we invite you to submit a revised version of the manuscript that addresses the points raised during the review process.

We look forward to receiving your revised manuscript.

Kind regards,

Guillaume Thierry, Ph.D.

Academic Editor

PLOS ONE

Journal Requirements:

3.Thank you for stating the following in the Acknowledgments Section of your manuscript:

"We thank the Australian Brain Foundation for a grant that helped fund this exploratory work."

5.We note that Figure(s) 1  in your submission contain copyrighted images. All PLOS content is published under the Creative Commons Attribution License (CC BY 4.0), which means that the manuscript, images, and Supporting Information files will be freely available online, and any third party is permitted to access, download, copy, distribute, and use these materials in any way, even commercially, with proper attribution. For more information, see our copyright guidelines: http://journals.plos.org/plosone/s/licenses-and-copyright.

a) You may seek permission from the original copyright holder of Figure(s) 1 to publish the content specifically under the CC BY 4.0 license.

Additional Editor Comments:

Dear Authors,

As you will see when reading the comments from the two reviewers who read your paper in great detail, your idea to test synesthetes with a colour-coded calculator is very appealing and has substantial potential. I agree with the reviewers that this is an excellent and intriguing idea, and this is the reason why I elected to take the editorial assignment for your submission. This being said, whilst Reviewer 2 is very positive about the paper in general and only has minor issues to raise, Reviewer 1 has found substantial weaknesses in the manuscript, with which I wholeheartedly agree. Reviewer 2 manage to be both concise and thorough, so I will not rewrite their comments here, but I urge you to consider every point made by this reviewer. I very much hope you will be willing to undertake the revisions required to bring the paper in line with the level of excellent excepted at PLoS. IN particular, I agree with Reviewer 2 that control groups are needed for exp 1 and 3, and I invite you to test control groups in earnest in order to address this shortcoming. Also, I invite you to provide access to the raw data and a full output of your statistical analyses so that the reviewers can verify that they agree with the way in which you report your results.

I very much look forward to seeing your revised paper, and I am convinced that this study will make an excellent contribution to the field once your manuscript is adequately revised.

Best regards,

Guillaume Thierry

Reviewers' comments:

Reviewer's Responses to Questions

**Comments to the Author**

1. Is the manuscript technically sound, and do the data support the conclusions?

Reviewer #1: Yes

Reviewer #2: No

2. Has the statistical analysis been performed appropriately and rigorously? 

Reviewer #1: Yes

Reviewer #2: I Don't Know

3. Have the authors made all data underlying the findings in their manuscript fully available?

Reviewer #1: Yes

Reviewer #2: No

4. Is the manuscript presented in an intelligible fashion and written in standard English?

Reviewer #1: Yes

Reviewer #2: Yes

5. Review Comments to the Author

Reviewer #1: The study explored the utility of a digital-color calculator (DCC) for people with grapheme-color synesthesia who sometimes may experience problems when the colors of numbers do not match their concurrent colors. The quantitative results showed an increase in calculation speed in the congruent when compared to the incongruent condition. The qualitative results were much more pronounced in that, most of the synesthetes experienced a decreased cognitive load (a sense of ease) in the congruent condition. Such quantitative and quantitative differences were not seen in the non-synesthete group. Although there has been a significant increase in the number of studies related to synesthesia, this is the first one to investigate the benefits of using a DCC. This is a well-done study and the DCC could be especially useful to many synesthetes who could be having trouble and some amount of frustration dealing with numbers in their everyday lives. This could also be beneficial to synesthetes with learning disabilities although its incidence in this population is unknown.

I understand that the authors were motivated by the case study SP who had a life changing experience with the introduction of DCC. However, the paper should include investigations where they have looked at the cognitive benefits of synesthesia (especially GC synesthesia). It is unsettling that such studies and theories such as the Dual-Coding theory have not been described in the introduction or the discussion sections. Some of these studies have shown only local effects (numbers, letters) but some others have shown more global effects (better vocabulary, creativity etc.). Is it possible that such cognitive advantages could be hindered by providing the non-learning-disabled synesthetes with such tools?

In the qualitative part of the results, I see that the synesthetes have used a lot of emotive language. These results must be explained/corroborated with findings from other investigations which have described the emotional attributes of synesthesia (see Perry & Henik, 2013, and Rouw & Scholte,2016 for example).

Finally, did the authors rule out cognitive, speech and language, and neurological impairments for the participants? Was any screening done (or at least asked for self-reports)?

Reviewer #2: Summary

The study investigates whether a digital calculator that represents digits in colors congruent or incongruent to the digit-color associations of grapheme-color synesthetes affects their speed of performing mathematical calculations.

Evaluation

I liked the idea of providing synesthetes a digital calculator that can be adjusted to represent their synesthetic color experiences. However, the study has many problems which prevent me from recommending it for publication in the present form. To address these problems would require a substantial rewrite.

First, on a theoretical level the claim that synesthesia “incurs an initial processing cost compared with its equivalent example from normal perception” is simply wrong. There are dozens of studies showing the automatic nature of the synesthetic experiences. Moreover, even the present study provides no evidence for this claim. If anything, the present study seems to suggest a *benefit* rather than a cost in situations in which the physical color concurs with the synesthetic experience. However, even this result is ambiguous (see below).

Second, the description of the method lacks important detail. For example, did the instructions emphasize speed? What exactly is represented by the dependent variable: is it the time from pressing the “next” key at the beginning of one calculation to pressing the “next” key for the next calculation? Were the “reading”, “working”, “transcribing” phases assessed separately? The information about the sample should be presented in the paper, not in the supplementary materials. In both, Experiment 1 and Experiment 3 a control group of non-synaesthetes is missing (in Experiment 2 there is a control group but this group is not matched to the synaesthtete group,which makes comparisons difficult). The missing control group is particularly critical in Experiment 3. It remains completely obscure whether the improvement in task speed is specific to synesthetes or whether this might also be found in non-synesthetes.

Third, for descriptive statistics it would be helpful to provide means and standard deviations in a Table. Regarding statistical analysis, it is not clear for me why the results are not compared by means of repeated measures or mixed model ANOVAs. What is the benefit of the modelling approach (BTW I have not found the analysis corresponding to Model 3 from S5 in the text, nor is there any justification for it)? Notably, the values in the text and the values in Figure 3 do not correspond. Specifically, from Figure 3A it is evident that the non-synesthetes show the same performance pattern across conditions as the synesthetes (i.e., congruent < neutral < incongruent). This latter result makes the conclusion that synesthetes have a specific advantage from using the colored digital calculator dubious.

Fourth, according to the PloSOne guidelines, raw data should be provided as part of the manuscript or its supporting information, or deposited to a public repository. This is not the case here.

6. PLOS authors have the option to publish the peer review history of their article (what does this mean?). If published, this will include your full peer review and any attached files.

Reviewer #1: **Yes: **Vijayachandra Ramachandra

Reviewer #2: No

---

## [Author Response · Author response to Decision Letter 0]

10 Mar 2021

Please see the attached Response to Reviewers document.

---

## [Decision Letter · Decision Letter 1]

14 May 2021

PONE-D-20-35946R1

Sharing the load: How a personally coloured calculator for grapheme-colour synaesthetes can reduce processing costs.

PLOS ONE

Dear Authors,

Thank you for the exchange of emails we had through the editorial office of PlosONE. As discussed in our exchange, I am now recommending acceptance of your paper for publication in PlosONE. Regarding the figures you sent me via the editorial office, I am very pleased to see them and I think the second figure (which features both the synaesthetes and control colour choices) is very demonstrative and striking. In order to allow for you to integrate that figure in the manuscript, I am making a minor revision decision here on the understanding that I will finalise acceptance upon receipt of your manuscript revision including the new figure.

Please refer to the figure in the main text and provide a detailed caption for it. Also, I would advise you to remove the white border around each colour patch sequence/column in the panel displaying the controls' colour choices, to make the two panels visually more directly comparable. Please also move the "Controls" label to the bottom of the figure and remove any unnecessary line / border. I am aware that you might find my guidance a little too specific here, but I really think this is good investment of your time as I predict that this figure will appeal to the readership and will raise interest.

Please submit the revised paper at your earliest convenience so that we can proceed to production.

Best regards,

Guillaume thierry

We look forward to receiving your revised manuscript.

Kind regards,

Guillaume Thierry, Ph.D.

Academic Editor

PLOS ONE

Journal Requirements:

Reviewers' comments:

Reviewer's Responses to Questions

**Comments to the Author**

1. If the authors have adequately addressed your comments raised in a previous round of review and you feel that this manuscript is now acceptable for publication, you may indicate that here to bypass the “Comments to the Author” section, enter your conflict of interest statement in the “Confidential to Editor” section, and submit your "Accept" recommendation.

Reviewer #1: All comments have been addressed

Reviewer #2: (No Response)

2. Is the manuscript technically sound, and do the data support the conclusions?

Reviewer #1: Yes

Reviewer #2: No

3. Has the statistical analysis been performed appropriately and rigorously? 

Reviewer #1: Yes

Reviewer #2: Yes

4. Have the authors made all data underlying the findings in their manuscript fully available?

Reviewer #1: Yes

Reviewer #2: Yes

5. Is the manuscript presented in an intelligible fashion and written in standard English?

Reviewer #1: Yes

Reviewer #2: Yes

6. Review Comments to the Author

Reviewer #1: In this revised version of the manuscript the authors have adequately addressed all my comments/suggestions. I have no further comments.

Reviewer #2: The authors have addressed some of the issues appropriately. However, two major concerns still remain.

They still claim that “synaesthesia “incurs at an initial cost”. They argue that the synaesthete brain needs additional processing paths to generate the synaesthetic experience. This claim is wrong. There is converging evidence that synaesthesia is characterized by hyper-connectivity in the brain, that is, there are direct pathways between colour and form areas that are not present in non-synaesthetes, see references below. Rather than due to *additional costly* processing, synesthesia is due to different wiring in the brain.

Hanggi J, Wotruba D, Jäncke L. Globally altered structural brain network topology in grapheme-color synesthesia. J Neurosci. 2011 Apr 13;31(15):5816-28. doi: 10.1523/JNEUROSCI.0964-10.2011.

Rouw R, Scholte HS. Increased structural connectivity in grapheme-color synesthesia. Nat Neurosci. 2007 Jun;10(6):792-7. doi: 10.1038/nn1906.

As noted in the previous review, it is necessary to include a control group of non-synaesthetes to test whether the improvement in task speed is specific to synesthetes or whether this might be found also in non-synesthetes. In order to evaluate the particular benefit of the Digital Colour Calculator this proof of evidence is mandatory.

7. PLOS authors have the option to publish the peer review history of their article (what does this mean?). If published, this will include your full peer review and any attached files.

Reviewer #1: **Yes: **Vijayachandra Ramachandra

Reviewer #2: No

---

## [Decision Letter · Decision Letter 2]

23 Aug 2021

PONE-D-20-35946R2

Sharing the load: How a personally coloured calculator for grapheme-colour synaesthetes can reduce processing costs.

PLOS ONE

Dear Dr. Berger,

Thank you for submitting your manuscript to PLOS ONE. After careful consideration, we feel that it has merit but does not fully meet PLOS ONE’s publication criteria as it currently stands. Therefore, we invite you to submit a revised version of the manuscript that addresses the points raised during the review process.

Thank you for submitting your manuscript to PLOS ONE. As you are aware, the original academic editor for this manuscript is unavailable. Since we were not able to secure a replacement academic editor, I have taken handling the manuscript. As part of my responsibilities in doing so, I must consider the review process throughout, including whether concerns raised throughout the review process have been addressed. As such, it was essential that the manuscript was thoroughly assessed by me. As the handling editor I am responsible for ensuring the submission meets PLOS ONE’s publication criteria, and it is not appropriate for me to solely rely upon the previous academic editors’ evaluation.

I understand that the previous academic editor may have indicated that the manuscript was ready for acceptance. However, no Accept decision was formally issued. Having performed my own evaluation of the submission I consider that a small number of straightforward revisions are necessary. I appreciate that this may be frustrating, especially if you had been previously advised the manuscript was ready for publication. I have not made the requests below without thoroughly assessing your manuscript, alongside the comments from the reviewers. These revisions are necessary for the manuscript to meet PLOS ONE’s publication criteria, which requires that conclusions are presented in an appropriate fashion and are supported by the data.

As you will be aware, Reviewer 2 has raised concerns throughout the review process relating to the study design and reported “cost” of processing in those with synesthesia. I address each of the concerns in turn below.

The main concern relating to the study is that studies 1 and 3 are missing non-synesthetic controls, and that the control participants in study 2 may not be matched to the synesthetic participants. Whilst I am sympathetic to these concerns, I do not think the concerns with the control group affect the manuscript’s conclusions. However, I do think that these concerns should be acknowledged and addressed in your limitations section.

The second concern noted relates to the reference to the “cost” of processing in synesthesia. I have read the discussion in lines 498-534, where the manuscript indicates exactly what is meant by “cost”. Specifically, it states that the additional information processing in the synesthetic brain has additional components, which use more energy than normal conscious experience. Whether this apparent additional cost is significant to synesthetes is not clear from this argument alone. Indeed, if such cost was meaningful, one would expect to see stronger evidence of a facilitation effect. However, I understand from your response-to-reviewers that the aim of the study was not to provide evidence of the additional cost of synesthetic processing. I was therefore surprised to in your conclusions that you refer to your “identification of the additional processing cost of synaesthesia”. This statement should be revised, since your manuscript does not present strong evidence of such an additional cost.

This relates to another limitation that should be better acknowledged. Whilst the manuscript presents evidence of congruency and interference effects, there is only moderate evidence to support a facilitation effect. Since it would be expected that most calculators available to synesthetes would most closely resemble the conditions of the *Control* condition (rather than incongruent condition, where larger effects were observed), there is only moderate evidence to support the real-world benefit of the DCC for synaesthetes, when considering response times.

I have provided a list of required changes to address these concerns, in addition to some minor changes that should be made. These changes are listed in order they appear in the manuscript, not in severity.

Lines 29-30: As noted above in your response-to-reviewers, the aim of the manuscript was not to show the additional cost of processing, and only limited evidence has been presented in this regard. The phrase “we introduce the additional processing cost of synaesthesia” must be revised in light of this concern.Line 124: Please note that colored text within the main body of the manuscript will be black when the manuscript is published. If you wish to demonstrate the colored digit, this should be included as part of a figure.The Methods section does not include a description of the task or what the different conditions are, although this information has been included in the Introduction. Since future readers will look to the Methods sections for these methodological details, I strongly encourage you to add this information to the Methods section.S1 and S2 files have not been included with the most recent versions of the submission – please ensure that these are included in your revised manuscript.The results of the survey are termed “qualitative results”. Qualitative data are non-numerical, so it is not appropriate to report the results of your Likert survey or any other numerical data under this heading (even if they are ordinal). My suggestion would be to rename these subheadings as “Questionnaire data” or similar.The limitations around the small facilitation effect and missing control condition need to be acknowledged. Specifically, in your limitations you should:Discuss whether the missing healthy controls from the first experiment impact the conclusions of the manuscript.Discuss whether the apparently different demographic characteristics of healthy participants in ‘Group Experiment: part 2’ impact the conclusions of the manuscript.Acknowledge that the real-world benefit to the wider synesthetic population is limited, given that only small effects were reported for the facilitation effect. I appreciate that greater facilitation effects are reported in the ‘Subsample Experiment: repeated measures’. However, since this population has specifically been selected for extreme responses to the congruent calculator, it is not appropriate to make conclusions about the wider synesthetic population based on the results of these participants.In your conclusion you should revise the statement “with our identification of the additional processing cost of synaesthesia”, since only limited effects were reported on this.

Thank you very much for your attention to these requests. I hope that they should be straightforward for you to complete, and this somewhat mitigates the understandable frustration of receiving another revision decision. I will assess your revisions when the manuscript is returned and if they are acceptable I will proceed to issue an Accept decision. I look forward to receiving your revised manuscript.

Kind regards,

George Vousden

Division Editor

PLOS ONE

Additional notes from the journal office:

Journal Requirements:

Reviewers' comments:

Reviewer's Responses to Questions

**Comments to the Author**

1. If the authors have adequately addressed your comments raised in a previous round of review and you feel that this manuscript is now acceptable for publication, you may indicate that here to bypass the “Comments to the Author” section, enter your conflict of interest statement in the “Confidential to Editor” section, and submit your "Accept" recommendation.

Reviewer #1: All comments have been addressed

Reviewer #2: (No Response)

2. Is the manuscript technically sound, and do the data support the conclusions?

Reviewer #1: Yes

Reviewer #2: No

3. Has the statistical analysis been performed appropriately and rigorously? 

Reviewer #1: Yes

Reviewer #2: Yes

4. Have the authors made all data underlying the findings in their manuscript fully available?

Reviewer #1: Yes

Reviewer #2: Yes

5. Is the manuscript presented in an intelligible fashion and written in standard English?

Reviewer #1: Yes

Reviewer #2: Yes

6. Review Comments to the Author

Reviewer #1: The authors have adequately addressed my comments. I have no further comments or suggestions for them.

Reviewer #2: (No Response)

7. PLOS authors have the option to publish the peer review history of their article (what does this mean?). If published, this will include your full peer review and any attached files.

Reviewer #1: **Yes: **Vijayachandra Ramachandra

Reviewer #2: No

---

## [Author Response · Author response to Decision Letter 2]

26 Aug 2021

Dear Dr Vousden,

Thank you for your careful reading of our manuscript along with the previous reviews, letters, decisions, and rebuttals. Your comments and recommendations are clear and very fair and we have happily attended to them as follows:

1. Lines 29-30: As noted above in your response-to-reviewers, the aim of the manuscript was not to show the additional cost of processing, and only limited evidence has been presented in this regard. The phrase “we introduce the additional processing cost of synaesthesia” must be revised in light of this concern.

Lines 28-30: this has been revised as requested (see tracked changes). We have removed reference to cost processing from the abstract.

2. Line 124: Please note that colored text within the main body of the manuscript will be black when the manuscript is published. If you wish to demonstrate the colored digit, this should be included as part of a figure.

The colored text within the main body has been removed.

3. The Methods section does not include a description of the task or what the different conditions are, although this information has been included in the Introduction. Since future readers will look to the Methods sections for these methodological details, I strongly encourage you to add this information to the Methods section.

We have included a description of the different conditions within the Methods as you have encouraged (lines 144 - 147).

4. S1 and S2 files have not been included with the most recent versions of the submission – please ensure that these are included in your revised manuscript.

All the supplementary figures are now uploaded with the most recent version of the submission.

5. The results of the survey are termed “qualitative results”. Qualitative data are non-numerical, so it is not appropriate to report the results of your Likert survey or any other numerical data under this heading (even if they are ordinal). My suggestion would be to rename these subheadings as “Questionnaire data” or similar.

We have changed the relevant subheadings and references in line with your suggestion to ‘Questionnaire data’.

6. The limitations around the small facilitation effect and missing control condition need to be acknowledged. Specifically, in your limitations you should:

a. Discuss whether the missing healthy controls from the first experiment impact the conclusions of the manuscript.

b. Discuss whether the apparently different demographic characteristics of healthy participants in ‘Group Experiment: part 2’ impact the conclusions of the manuscript.

c. Acknowledge that the real-world benefit to the wider synesthetic population is limited, given that only small effects were reported for the facilitation effect. I appreciate that greater facilitation effects are reported in the ‘Subsample Experiment: repeated measures’. However, since this population has specifically been selected for extreme responses to the congruent calculator, it is not appropriate to make conclusions about the wider synesthetic population based on the results of these participants.

In response to you subpoint (a) we have discussed why the issue of the ‘missing healthy controls’ from part 1 of the first experiment does not affect our conclusion (line 595 – 600).

Similarly, in response to (b) we have discussed why the dissimilar demographics of the additional control group do not affect our conclusions (line 592 - 603). 

Finally, regarding subpoint (c), we have added a formal acknowledgement that the real-world benefit to the wider synaesthete population is likely to be more limited (line 496 - 500).

7. In your conclusion you should revise the statement “with our identification of the additional processing cost of synaesthesia”, since only limited effects were reported on this.

We have revised the final concluding statement, making our conclusions regarding the reduction of the processing costs of synaesthesia more tentative (line 568 – 570).

We look forward to hearing your foreshadowed final decision for this manuscript, so that it may proceed to publication and be read by the wider science community.

Yours sincerely,

Dr Joshua Berger on behalf of the authors.

---

## [Editor Report · Decision Letter 3]

9 Sep 2021

Sharing the load: How a personally coloured calculator for grapheme-colour synaesthetes can reduce processing costs.

PONE-D-20-35946R3

Dear Dr. Berger,

We’re pleased to inform you that your manuscript has been judged scientifically suitable for publication and will be formally accepted for publication once it meets all outstanding technical requirements.

Kind regards,

George Vousden

Division Editor

PLOS ONE
---

## [Editor Report · Acceptance letter]

13 Sep 2021

PONE-D-20-35946R3 

Sharing the load: How a personally coloured calculator for grapheme-colour synaesthetes can reduce processing costs. 

Dear Dr. Berger:

I'm pleased to inform you that your manuscript has been deemed suitable for publication in PLOS ONE. Congratulations! Your manuscript is now with our production department. 

Kind regards, 

on behalf of

Dr. George Vousden 

Staff Editor

PLOS ONE